# Expression Pattern of T-Type Ca^2+^ Channels in Cerebellar Purkinje Cells after VEGF Treatment

**DOI:** 10.3390/cells10092277

**Published:** 2021-09-01

**Authors:** Jonas Tjaden, Annika Eickhoff, Sarah Stahlke, Julian Gehmeyr, Matthias Vorgerd, Verena Theis, Veronika Matschke, Carsten Theiss

**Affiliations:** 1Department of Cytology, Institute of Anatomy, Ruhr-University Bochum, Universitätsstr. 150, 44801 Bochum, Germany; jonas.tjaden@rub.de (J.T.); annieickhoff@hotmail.de (A.E.); Sarah.stahlke@rub.de (S.S.); julgeh@t-online.de (J.G.); Verena.theis@rub.de (V.T.); veronika.matschke@rub.de (V.M.); 2Department of Neurology, Neuromuscular Center Ruhrgebiet, University Hospital Bergmannsheil, Ruhr-University Bochum, Buerkle-de-la-Camp-Platz 1, 44789 Bochum, Germany; matthias.vorgerd@bergmannsheil.de

**Keywords:** Purkinje cell, VEGF, *Kdr*, T-type Ca^2+^ channels, *Cacna1g*, *Cacna1h*, *Cacna1i*

## Abstract

T-type Ca^2+^ channels, generating low threshold calcium influx in neurons, play a crucial role in the function of neuronal networks and their plasticity. To further investigate their role in the complex field of research in plasticity of neurons on a molecular level, this study aimed to analyse the impact of the vascular endothelial growth factor (VEGF) on these channels. VEGF, known as a player in vasculogenesis, also shows potent influence in the central nervous system, where it elicits neuronal growth. To investigate the influence of VEGF on the three T-type Ca^2+^ channel isoforms, Cav3.1 (encoded by *Cacna1g*), Cav3.2 (encoded by *Cacna1h*), and Cav3.3 (encoded by *Cacna1i*), lasermicrodissection of in vivo-grown Purkinje cells (PCs) was performed, gene expression was analysed via qPCR and compared to in vitro-grown PCs. We investigated the VEGF receptor composition of in vivo- and in vitro-grown PCs and underlined the importance of VEGF receptor 2 for PCs. Furthermore, we performed immunostaining of T-type Ca^2+^ channels with in vivo- and in vitro-grown PCs and showed the distribution of T-type Ca^2+^ channel expression during PC development. Overall, our findings provide the first evidence that the mRNA expression of Cav3.1, Cav3.2, and Cav3.3 increases due to VEGF stimulation, which indicates an impact of VEGF on neuronal plasticity.

## 1. Introduction

The vascular endothelial growth factor (VEGF) was first discovered in 1983, where it was responsible for the massive ascites of peritoneal tumors by increasing vascular permeability [1]. In the following years, more important functions of VEGF, like neovascularisation during developmental processes of organs and pathologies, and even crucial roles in the central nervous system (CNS) and peripheral nervous system (PNS) independent from endothelial stimulation were discovered [2,3,4]. In mammals, VEGF A, B, C, D and the placental growth factor are known VEGF isoforms and associated with the VEGF receptors VEGFR-1 (encoded by *Flt1*), VEGFR-2 (encoded by *Kdr*), VEGFR-3 (encoded by *Flt4*), and the two co-receptors neuropilin-1 (encoded by *Nrp1*) and neuropilin-2 (encoded by *Nrp2*) [5,6]. There are several processes that can be triggered by VEGF in the CNS. These include neuroregeneration, neuroprotection, neurogenesis, and axonal outgrowth [7,8,9]. To investigate the impact of VEGF on neuronal cells and their plasticity, Purkinje cells (PCs) are an excellent model. They are crucial for the function of the cerebellum, easy to identify due to their unique appearance, and have a distinct neuronal circuit [10,11]. It is known that VEGF induces axonal outgrowth and dendritogenesis in juvenile PCs, whereas the underlying molecular mechanisms and functional aspects remain mostly uncertain [9,12]. Reorganisation of neuronal structures in addition with a functional change, for example, of ion channel density or function, is called neuronal or synaptic plasticity. This, in turn, is the cellular and molecular equivalent of memorization and learning [13]. Yet, it is not fully understood which extra- and intracellular mechanisms contribute to synaptic plasticity, and there are many different pathways considered to influence this reorganisation [14,15]. In PCs, Ca^2+^ ions play a key role in the processes of synaptic plasticity [16]. Intracellular Ca^2+^ ion concentration, which affects several enzymes and controls different signaling pathways, has long been the focus of research on synaptic plasticity [17,18]. Voltage-gated Ca^2+^ channels, the high-voltage-gated P/Q-type channels and the low-voltage-gated T-type Ca^2+^ channels, intracellular calcium stores and endogenous Ca^2+^ buffer (ECB) are responsible for the Ca^2+^ ion homeostasis in PCs [19,20,21]. Recent studies took the barely known low-voltage-gated T-type Ca^2+^ channels into account to explain synaptic plasticity in PCs [22,23,24,25,26]. These T-type Ca^2+^ channels can be classified into three subtypes, namely, Cav3.1 (encoded by *Cacna1g*), Cav3.2 (encoded by *Cacna1h*), and Cav3.3 (encoded by *Cacna1i*), and their expression in PCs was proven lately [21,27]. Furthermore, these channels, which can depolarize from a hyperpolarized state, contribute to excitatory postsynaptic potentials (EPSP) and complex spikes in PCs [28,29]. Especially, the long-lasting Ca^2+^ influx can evoke a significantly high intracellular Ca^2+^ concentration and therefore influence the way of signal conduction and contribute to synaptic plasticity [30,31]. To extend the knowledge of the effect of VEGF, we wanted to examine its role in neuronal functioning and synaptic plasticity and therefore analysed the expression of *Cacna1g*, *Cacna1h*, and *Cacna1i* after VEGF treatment.

## 2. Materials and Methods

All procedures were carried out under established standards of the German federal state of North Rhine Westphalia, in accordance with the European Communities Council Directive 2010/63/EU on the protection of animals used for scientific purpose.

### 2.1. Enriched PC Culture

The procedure to obtain a highly pure PC culture was already described before [32]. Briefly, anti-GD3 ganglioside antibodies were obtained with the help of a R24 hybridoma cell line (hybridoma R24, HB-8445, ATCC, Manassas, VA, USA). On day two, eight petri dishes for the immunopanning procedure (351007, Corning Falcon, Corning, NY, USA) were prepared with 3 mL of 50 mM Tris-HCl (pH 9.5), 24 µL of goat anti-mouse IgG antibody (31160, Thermo Fisher, Waltham, MA, USA), and 60 µL goat anti-mouse IgM antibody (ab9167, Abcam, Cambridge, UK). On day three, prepared petri dishes were washed with filtered phosphate-buffered saline (PBS, pH 7.4), and the anti-GD3 supernatant as well as the anti-Thy1.1 antibody were added.

Cerebella were obtained from male and female Wistar rat pups of p0. Digesting of the tissue was performed with a sterile filtered solution including: 5 mL Earle’s solution with EDTA and NaHCO3, 15 µL of DNase (5 mg/mL), 1.2 mg of L-Cysteine (168149-2.5G, Sigma-Aldrich, St. Louis, MO, USA), 60 µL of papain solution (31.43 mg/mL, PAPL, Worthington LS 003119 in 60 µL of papain activation solution (6.66 mg L-Cystein, 44 µL 0.25 M EDTA solution, 47.16 µL Mercaptoethanol in 10 mL of ddH_2_O)). Dissected cerebella were incubated in this solution for 1 h at 35.5 °C. After incubation, cerebella were triturated and a centrifugation step with Percoll gradients was performed, as described before [32].

Afterwards, the PCs were isolated with the help of anti-GD3 and anti-Thy1.1 antibodies. The cells were counted using a Neubauer chamber, and 100.000–120.000 cells were plated in 96er well plates in serum-free medium. Serum-free medium was replaced the day after extraction and then every second day. VEGF (V4512, Sigma-Aldrich, c = 0.1 μg/mL in ddH_2_O) was mixed with serum-free medium and applied to the group of stimulated PCs.

### 2.2. Laser Microdissection

Dissected cerebella were frozen using isopentane cooled by liquid nitrogen and stored at −80 °C. Removable parts of the cryostat were cleaned with a solution containing 1 mM EDTA and 0.1 M NaOH in DEPC-treated water to inhibit RNases. After fixation with tissue freezing medium (No. 14020108926, Leica, Wetzlar, Germany), 12 µm sections were obtained with the help of the cryostat (Leica microsystems CM3050 S, Wetzlar, Germany). Sections were mounted onto RNase-free polyethylene naphthalate-(PEN)-membrane slides manufactured for LMD (No. 11505151, Leica) and dried on a heater at 40 °C for at least 20 min. Tissues were stained by applying a solution containing 1% methylene blue, 1% azure II, and 1% Borax in DEPC-treated water and washing with DEPC afterwards, as described by Pieczora et al. [33]. Slides were microdissected using the LMD6500 system (Leica Microsystems, Wetzlar, Germany).

Then, 1,000,000 μm^2^ of enriched PCs, which equals 2000 PCs, and 1,000,000 μm^2^ of the molecular layer from 20 female and male Wistar rats were isolated in total and samples were collected in 20 μL of lysis solution (AM1931, Invitrogen, Waltham, MA, USA) each. All samples were stored at −80 °C.

### 2.3. RT-qPCR

For total RNA isolation from cultured cells, NucleoSpin^®^ miRNA isolation kit (No. 740304, Macherey-Nagel, Düren, Germany) was used according to the manufacturer’s protocol. Briefly, pelleted cells were resuspended in 300 µL ML lysis buffer. Then, 100% EtOH was added, and samples were loaded onto the NucleoSpin RNA Column and centrifuged briefly. After washing with different buffers and incubation with rDNase, the RNA columns were washed with buffer MW 1 and MW 2 and then the total RNA was eluted in 30 µL of nuclease-free water and stored at −80 °C.

For the isolation of the total mRNA of the LCM samples, we used a RNAqueous-Micro Kit for LCM samples (AM1931, Invitrogen) according to the manufacturer’s protocol. Briefly, wash solutions were prepared, and the elution solution was heated to 95 °C. Dissected PC samples were stored in lysis solution at −80 °C, and 100 µL of lysis solution containing the cells were incubated, and to recover large and small RNAs, 129 µL of 100% ETOH were added. After three washing steps, the RNA was eluted by using 10 µL of preheated elution solution. To inactivate DNases, the DNase protocol of the RNAqueous-Micro Kit for LCM samples was performed after the RNA extraction process. In the end, the RNA was stored at −80 °C.

cDNA was synthesised using qScript cDNA SuperMix (95048-100, Quantabio, Beverly, MA, USA). Therefore, 20 µL samples consisting of 10 µL undiluted total RNA, 4 µL qScript cDNA SuperMix and 6 µL RNAse-free water were incubated for 5 min at 25 °C, 30 min at 42 °C, 5 min at 85 °C and then cooled down to 4 °C. The samples were stored at −20 °C.

Quantitative PCR was performed with the aid of GoTaq^®^ qPCR Master Mix (No. A6001, Promega, Madison, WI, USA). The amount of cDNA was adjusted to 50 ng/µL for the quantification of culture purity and to 200 ng/µL for the examination of the T-type Ca^2+^ channel mRNA. All analyses were done in triplicate. The qPCR was conducted according to the manufacturer’s protocol. The following primers were used: *Cacna1g* (5′-GTC ATT TGC TGT GCC TTC TTC-3′, 5′-TGT TAG TGA TGT TCC TGG TGT C-3′, Microsynth, Balgach, Switzerland), *Cacna1h* (5′-CTT CAT CTT CGG CAT TGT TGG-3′, 5′-CCT CCT CCG TCT GGT AGT AT-3′, Microsynth), *Cacna1i* (5′-AGC CTG TCA CTC ACA TCT CT-3′, 5′-TAC TGC TGA ACT TCC TGG CT-3′, Microsynth), *Calb1* (5′-GAA GGA AAG GAG CTG CAG AA-3′, 5′-TCT GCC CAT ATT GAT CCACAA A-3′, Microsynth), *Gfap* (5′-GAG TGG TAT CGG TCC AAG TTT-3′, 5′-TTG GCG ATA GTC ATT AG-3′, Microsynth), *Rbfox3* (5′-GAG AAG CTC AAT GGG ACG AT-3′, 5′-CAT ATG GGT TCC CAG GCT-3′, Microsynth), *Tnfα* (5′-GCT CCC TCT CAT CAG TTC CA-3′, 5′-GCTACG GGC TTG TCA CTC-3′, Microsynth), *Gabra6* (5′-CCG ATG AGA CTG GTT AAC TTC C-3′, 5′-TCT TCT GGG ACC TCT ACT GAA T-3′, Microsynth), *VGlut1* (5′-CCT GCG CAG TCG TCA TAT AAT-3′, 5′-GCC CTT GGA GTG TGA GTA TC-3′, Microsynth), *Flt1* (5′-GTG AAG AGT GGG TCG TCA TTC-3′, 5′-CTA TGG TTT CCT GCA CCT GTT-3′, Microsynth), *Kdr* (5′-TCC CAG AGT GGT TGG AAA TG-3′, 5′-ACT GAC AGA GGC GAT GAA TG-3′, Microsynth), *Flt4* (5′-CTG GAC ACC CTG TAA GAC ATT T-3′, 5′-AGT GGT CAC CTC CTT CCA-3′, Microsynth), *Nrp1* (5′-AGA TCG CCT ACA GTA ACA ATG G-3′, 5′-CTT GTG GAG AGA GGT GTA AAG G-3′, Microsynth), *Nrp2* (5′-GGC TTC TCA GCA CGT TAC TAT T-3′, 5′-TGA GGC ACT GAT CTG TTC ATT AG-3′), Glyceraldehyde-3-phosphate dehydrogenase (*Gapdh*) (5′-ACT CCC ATT CTT CCA CCT TTG-3′, 5′-CCC TGT TGC TGT AGC CAT ATT-3′, Microsynth).

The 2^−ΔΔct^ method [34] was used to analyse the relative mRNA expression in the enriched PC culture and the samples obtained with the help of LMD. Therefore, all ct-values of the examined mRNA were normalized against the ct-values of *Gapdh*. An unpaired two-tailed *t*-test was performed to prove statistical significance between the relative expression levels of the mRNAs. Microsoft Excel and GraphPad Prism 5 were used for statistical analysis.

### 2.4. Immunostaining Experiments

For the examination of the in vitro-grown PCs, 200,000 cells were seeded per poly-D-lysin-coated cover slip and then fixed after 48 h of cultivation in 4% paraformaldehyde. For visualisation of the T-type Ca^2+^ channels in in vivo-grown PCs, 12 µm sections of rat cerebellum were obtained using a cryostat (Leica microsystems CM3050 S, Germany). Therefore, the tissue was embedded in tissue freezing medium (No. 14020108926, Leica) and sections were applied to SuperFrost Plus Adhesion slides (Thermo Scientific, J1800AMNZ). After washing three times with PBS (each 10 min), cells and cryosections were permeabilized with Triton (T8532, Sigma-Aldrich), and unspecific binding was blocked with 2% goat serum for the cells or 10% goat serum for the cryosections (G9023, Sigma-Aldrich) for 20 min (cells) or 30 min (cryosections). To stain PCs, samples were incubated with primary anti-calbindin antibodies (mouse, 1:600, C 9848, Sigma-Aldrich) at 4 °C overnight. On the next day, the samples were washed with PBS three times for 10 min and incubated with secondary anti-mouse TRITC antibody (goat, 1:1000 T5393, Sigma-Aldrich) at room temperature for 2 h, respectively, 2.5 h. After additional washing, the second primary antibody anti-Cav3.1 (rabbit, 1:1500 ACC21, Alomone Labs), anti-Cav3.2 (Rabbit, 1:1500 ACC25, Alomone Labs) or anti-Cav3.3 (rabbit, 1:1000 ACC-009, Alomone Labs) was applied at 4 °C overnight. Unattached antibodies were washed with PBS, and secondary anti-rabbit antibodies (donkey, AF488, Lifetech A21206, 1:1000 in PBS) were applied at room temperature for 2 h, respectively, 2.5 h. Lastly, nuclear counter staining was performed with Hoechst 33342 trihydrochloride (1:1000 in PBS, B2261, Sigma-Aldrich) for 20 min, followed by washing with PBS and embedding the samples with Fluorshield™ (F6937, Sigma-Aldrich). As negative controls, staining using only secondary antibody were done.

## 3. Results

In order to investigate the effect of VEGF on the mRNA expression of T-type Ca^2+^ channels, we incubated the PC culture with VEGF and analysed potential changes with the aid of RT-qPCR. Additionally, we examined the common characteristics and the composition of T-type Ca^2+^ channels in PCs during development in vivo and in vitro, via LMD in combination with RT-qPCR as well as immunostaining.

### 3.1. Accuracy of LMD

With the help of LMD (Figure 1A), we examined mRNA expression patterns of specific cell markers to ensure precise dissection of PCs (Figure 1B–E). At stage p9, the cerebellar cortex consists of four layers: the external granular cell layer (eGCL), the molecular layer (ML), the PC layer (PCL), and the internal granular cell layer (iGCL). In methylene blue-stained cryosections, these layers, and especially the PCs, are easy to dissect via LMD (Figure 1A). Nevertheless, methylene blue-staining of sections from p0 rat cerebella is not sufficient to detect PCs. The mRNA of the dissected PCs of both developmental stages p9 and p30 was then analysed and quantified. Calbindin mRNA (*Calb1*) is a specific marker of PCs and was predominant in LMD samples of microdissected PCs (p9 and p30) (1.002 ± 0.0043; 1.001 ± 0.0039) (Figure 1B,C). Other cell markers such as for glia cells (glial fibrillary acidic protein, *Gfap*) and neurons other than PCs were only weakly expressed in both samples (p9 and p30) of microdissected PCs (p9: *Gfap*: 0.1842 ± 0.0250; *Rbfox3*: 0.0405 ± 0.0060; *Gabra6*: 0.1916 ± 0.0161, *VGlut1*: 0.0380 ± 0.00078; p30: *Gfap*: 0.0517 ± 0.0059, *Rbfox3*: 0.0695 ± 0.0052; *Gabra6*: 0.1916 ± 0.0169; *VGlut1*: 0.2288 ± 0.0314). Here, NeuN (*Rbfox3*) is a universal marker for neuronal cells in the central nervous system (CNS) excluding PCs [35]. α6-GABA(A) receptor mRNA (*Gabra6*) is a marker for cerebellar granule cells [36] and Vglut1 mRNA (*Vglut1*) is specific for glutamatergic neurons [37]. To further assess these results, we additionally dissected and analysed the ML. In the ML, *Gfap* was highly expressed (ML9: *Gfap*: 9.299 ± 0.9863, ML30: *Gfap*: 15.01 ± 1.127) and the other cell markers, including *Calb1*, had a significantly lower expression (ML9: *Calb1*: 1.020 ± 0.1494; *Rbfox3*: 1.507 ± 0.1006; *Gabra6*: 0.03597 ± 0.00643; *Vglut1*: 1.686 ± 0.2556; ML30: *Calb1*: 1.107 ± 0.3103; *Rbfox3*: 1.145 ± 0.1680; *Gabra6*: 1.109 ± 0.2166; *Vglut1*: 0.6062 ± 0.1668) (Figure 1D,E).

### 3.2. PC Culture and VEGF Receptor Expression

To further visualise the PC culture, we performed immunostaining with an antibody against calbindin, particularly concentrated in the dendrites and perikarya of cerebellar PCs (Figure 2A). With the help of a phase-contrast microscope, cultures were analysed in terms of vitality (Figure 2B). The results indicated PCs with growing axons or dendrites. Additionally, we quantified mRNA expression levels of *Flt1*, *Kdr*, *Flt4*, *Nrp1*, and *Nrp2* in purified PC cultures (Figure 2C) and compared this to LMD samples of PCs (Figure 2D). Indeed, expression of *Kdr* was strongest in purified PC cultures as well as in LDM samples from PCs of p9 and p30 rat cerebella compared to all other receptors.

### 3.3. Age-Dependent mRNA Expression of T-Type Ca^2+^ Channels

Next, we examined the mRNA expression pattern of T-type Ca^2+^ channels in juvenile (p9) and adult (p30) PCs. The results showed that *Cacna1g*, *Cacna1h*, and *Cacna1i* were expressed at both age stages (p9 and p30) (Figure 3).

### 3.4. Immunostaining of T-Type Ca^2+^ Channel

In order to further analyse T-type Ca^2+^ channel distribution during development and in our purified PC culture, cryosections of p0 and p9 rat cerebella and isolated PCs from the culture were stained with specific antibodies to Cav3.1, Cav3.2, and Cav.3.3 in combination with anti-calbindin antibodies. p0 and p9 rat cerebella have a typical cortical arrangement in layers. Starting from the inside the internal granular cell layer (iGCL), the PC layer (PCL), the molecular layer (ML), and the external granular cell layer (eGCL) of the cerebellar cortex can be distinguished.

#### 3.4.1. Immunostaining with Anti-Cav3.1 Antibody

The immunohistochemical staining demonstrated a clear colocalization of PCs (calbindin-positive) and the T-type Ca^2+^ channel. In the PC culture, PCs (red) with developing dendrites or axons were visible. Cav3.1 was expressed along the soma and the dendrites or axons (Figure 4A–D). In neonatal cryosection (p0), calbindin-positive PCs (red) showed a diffuse Cav3.1 expression (green) around them, but also along their dendrites (Figure 4E–H). PCs of p9 revealed a prominent dendritic outgrowth into the ML, with Cav3.1 located mainly at the dendrites (Figure 4I–L).

#### 3.4.2. Immunostaining with Anti-Cav3.2 Antibody

The PCs of the PC culture had developing dendrites or axons, which connected among each other. Here, Cav3.2 seemed to be located along these cell extensions (Figure 5A–D). The Cav3.2 signal of the p0 cryosection was prominent around the PC somata and the dendrites (Figure 5E–H). In the p9 cryosections, calbindin-positive PCs (red) showed developing dendrites, along with a clearly visible eGCL. At this stage, a clear signal in the PCL and ML was detectable but, compared to p0 PCs, the signal was weaker (Figure 5I–L).

#### 3.4.3. Immunostaining with Anti-Cav3.3 Antibody

The results demonstrated a clear colocalization of Cav3.3 and PCs. The cultured PCs expressed Cav3.3 mainly at the soma (Figure 6A–D). In cryosections of p0 PCs, a lower signal was detected around the PCs and along the dendrites (Figure 6E–H). The in vivo-grown PCs of age p9 showed a strong signal in the ML, where the PC dendrites were located (Figure 6I–L).

### 3.5. Effect of VEGF on Cacna1g, Cacna1h and Cacna1i

After we evaluated the PCs of the PC culture and their T-type Ca^2+^ channel expression and compared the results with the expression in in vivo developed PCs, we examined the effect of VEGF on the expression of these channels in PCs. Therefore, the expression of *Cacna1g*, *Cacna1h*, and *Cacna1i* in cultured PCs was analysed subsequent to 24 h and 48 h of VEGF stimulation. The data showed a predominantly significant upregulating impact of VEGF on the mRNA levels of the T-type Ca^2+^ channels (Figure 7). *Cacna1g* was significantly upregulated after 24 h and 48 h stimulation with VEGF. In contrast, *Cacna1h* expression showed no difference after 24 h VEGF incubation, but a significant increase after 48 h VEGF treatment, whereas *Cacna1i* levels increased after 24 h VEGF incubation and were not significantly different after 48 h VEGF treatment.

## 4. Discussion

VEGF is known for its role in cancerogenesis and also for its neuroprotective effects [2,3,4]. The understanding of its explicit subcellular effects on neurons and influence on neuronal functioning is the subject of current research. In this context, our study aimed to examine the effect of VEGF on cerebellar PCs and their T-type Ca^2+^ channel expression. Therefore, we established an up-to-date method to cultivate PCs and compared the in vitro and in vivo expression of these channels and analysed changes during PC development.

### 4.1. Comparability of In Vivo- and In Vitro-Grown PC

The first task of our experiments was to investigate the comparability of dissected PCs samples with enriched PC cultures [32]. Using LMD, we examined defined samples of cerebellar tissue [38]. For this purpose, we dissected stained PC somata from the PCL as well as sections of the ML of p9 and p30 rat cerebella. In general, the parallel fibers (PF) of the granule cells, climbing fibers (CF) from the inferior olive, and glial cells surround PCs. The specific analysis of the cell composition of the respective samples revealed that a high amount of *Calb1* was present in the PC samples and a high amount of *Gfap* was present in the dissected ML samples. *Calb1* is a specific marker for PCs [39], while *Gfap* acts as a specific marker for glial cells [40]. *Rbfox3* as a marker for other cerebellar neurons than PCs, *Gabra6* as a marker for cerebellar granular cells, and *Vglut1* as a marker for glutamatergic neurons were only very weakly expressed in the PC samples [35,36,37]. This demonstrates that we had excised PCs precisely using LMD, which was of course ideal for further studies specific to PCs.

In the ML, there are glial cells, GABAergic stellate and basket cells and the dendrites of PCs, granular cells (parallel fibers) and neurons from the inferior olive (climbing fibers) [41]. The analysis of the dissected ML also reflects this very well. We were able to detect a high amount of *Gfap* in the samples compared to the other markers. This means that the mRNA markers were mostly expressed at the somata and not in the dendrites, like *Gabra6* in granular cells. These results demonstrated the specificity and accuracy of the LMD method to analyse specifically defined cells and tissue areas. The comparability with the PC culture is underlined by the experiments performed by Tjaden et al. 2018. Here, the cell composition of the PC culture was analysed after performing the purification protocol. There, the markers *Calb1*, *Gfap*, *Rbfox3*, *Tnfa*, *Gabra6*, and *Vglut1* were used [32]. These results showed a high amount of *Calb1* compared to all other markers. Unfortunately, the staining to identify PCs in the cryosections before performing LMD is only possible with cerebella older than p9, because staining and identification of younger PCs is insufficient. Nevertheless, this analysis of the specificity of the purification method for the isolation and cultivation of PCs showed an equivalent result compared to the PC samples, suggesting a high similarity.

In addition, our data revealed the mRNA expression pattern of different VEGF receptors and co-receptors. *Flt1*, *Kdr*, *Flt4, Nrp1*, and *Nrp2* were expressed in PCs in vivo and in vitro. The importance and function of the co-receptors known for their role in cancerogenesis remains unclear with respect to the central nervous system and the cerebellum. In cancer stem cells, VEGF and neuropilin receptors contribute to important aspects of tumorgenesis, self-renewal, and cancer cell survival [42]. Regarding the cerebellum, studies have shown that VEGFR-2 in particular exerts a crucial role in mediating VEGF signaling during PC development in young age stages [9]. We also showed that *Kdr* is expressed in in vivo- and in vitro-grown PCs and that a decrease in *Kdr* expression with increasing maturation stage of the in vivo-gown PCs could be detected. The high similarity between the dissected PCs and the PC culture indicates that VEGF most likely mediates its effect via VEGFR-2 in the in vivo- and in vitro-grown PCs. This very good comparability of the two study samples is the prerequisite for further parallel discussion of the expression patterns of Cav3.1, Cav3.2, and Cav3.3 in dissected and cultured PCs and the influence of VEGF on these cells.

### 4.2. Expression of T-Type Ca^2+^ Channels in PCs

T-type Ca^2+^ channels are low-voltage-gated calcium channels that depolarise near the resting membrane potential and play a critical role in pulsatile excitation and calcium homeostasis in neurons [43]. To date, three subtypes, Cav3.1, Cav3.2, and Cav3.3, and several splice variants have been described [44,45,46]. Particularly in the field of synaptic plasticity, whether NMDA receptor-dependent or -independent, these channels represent a key point in research [22,47,48,49]. The results of our research showed for the first time an increase in the relative mRNA expression of *Cacna1g* and *Cacna1i* and a decrease in the relative mRNA expression of *Cacna1h* during growth and ageing (p9 to p30) in PCs. Consistently, it has already been described that, in forebrain neurons, low-threshold calcium currents increase during ageing due to synaptic plasticity [50]. Though, results from retinal neurons show an increase in Cav3.2 currents during the critical phase of synaptogenesis and a decrease in adult mice [51]. Maybe that is the reason for the significant decrease of *Cacna1h* in adult PCs. Interestingly, the electrophysiological characteristics of Cav3.2 differ from Cav3.1 and Cav3.3, in the way that Cav3.2 has a longer lasting current than Cav3.3, but activates and inactivates slower than Cav3.1 [52]. The decrease of *Cacna1h* suggests a subordinate role of Cav3.2 in neuronal functioning of adult PCs.

In cerebellar PCs, signal conduction occurs by receiving mainly excitatory glutamatergic input from granule cells via PF as well as CF. At the PF–PC synapse, activation of the postsynaptic membrane generates excitatory postsynaptic potentials that require spatial and temporal summation to result in sufficient signal conduction. At the CF–PC synapse, activation by α-amino-3-hydroxy-5-methyl-4-isoxazolepropionic acid (AMPA) receptors generates so-called “complex spikes”. The term “complex spike” refers to a depolarisation that is triggered by AMPA receptors and subsequently spreads through many dendrites via P/Q- and T-type Ca^2+^ channels [20]. Much of the depolarisation is driven by Ca^2+^ ions and this calcium influx is thought to be conducted through T-type Ca^2+^ channels [53]. The evoked low-threshold Ca^2+^ ion influx can, on the one hand, cause further depolarisation of the postsynaptic membrane and, on the other hand, activate enzymes that bring about various intracellular mechanisms [54]. Ultimately, this intracellular enzyme cascade contributes to synaptic plasticity [55,56]. In this context, it is important to further analyse the distribution and temporal and regional expression of T-type Ca^2+^ channels. During maturation, Cav3.1 and Cav.3.3 showed an increase in staining in the PCL and ML, where the functional channels were predominantly localised along the PC dendrites, while the Cav3.2 signal remains uniform. Especially, an increase of staining of Cav3.1 and Cav3.3 could be detected at the dendrites. In p0 PCs, the Cav3.2 signal was localised at the soma of the PCs. Immunostaining of the PC culture showed PCs with developing dendrites and axons that also interconnect. The strong staining of the three T-type Ca^2+^ channels compared to p0 cryosections possibly indicates a pre-ageing process that continues during cultivation. Other work showed consistent findings of T-type Ca^2+^ channels distribution and expression in the cerebellum, but only in the adult stage after maturation. It should be emphasised that, in the study by McKay et al. (2006), Cav3.3 protein was found in the dendrites of rat PCs, whereas in the experiments of Talley et al. (1999) *Cacna1g* was the strongest expressed mRNA. The immunostaining results of Molineux et al. (2006) showed a strong signal of Cav3.3 in the dendrites, a clear signal of Cav3.2 near the PC somata, and a slight signal of Cav3.1 in adult PCs. However, in the review by Isope et al. (2012), the authors conclude that there is strong expression of Cav3.1 and less strong expression of Cav3.2 and Cav3.3 in PCs [27,28,57,58,59]. Our results showed a consistent expression compared to the studies above, and it seems that Cav3.1 and Cav3.3 are more important to PCs than Cav3.2. Furthermore, the results underlined a comparable expression of the three T-type Ca^2+^ channels between the PC culture and the in vivo-grown PCs. A recent paper describes the changes of Cav3.1 and Cav3.2 during development in mouse brains. In mouse cerebella, Cav3.1 and Cav3.2 expression increased during development, with a much smaller increase of Cav3.2 [21]. Our results underline the strong and increasing expression of Cav3.1 during development and a uniform signal of Cav3.2. Especially in in vivo p0 PCs, the signal of Cav3.2 is located at the soma. Overall, our data extend the understanding of T-type Ca^2+^ channel expression and highlight a change during cerebellar maturation with a focus on PCs. Furthermore, this process of change in T-type Ca^2+^ channels expression may contribute to the processes involved in synaptic plasticity.

### 4.3. Impact of VEGF on T-Type Ca^2+^ Channels Expression

To further investigate the impact of VEGF on the mRNA expression of T-type Ca^2+^ channels, we performed RT-qPCR experiments with the PC culture after VEGF treatment. In a larger context, this analysis provides the basis to further investigate VEGF’s role in neuronal plasticity and signal conduction [22,23]. While VEGF has been described as a neurotrophic and neuroprotective factor in CNS and PNS, triggering neuronal growth among others, not much is known about its influence on functional aspects such as neuronal signaling [8,9,60,61]. Our analysis showed a significant increase of T-type Ca^2+^ channel gene expression in PC cultures due to VEGF treatment. These results extend the current knowledge of the effects of VEGF. The present data show for the first time a highly significant increase of *Cacna1g* after 24 h and 48 h incubation of PCs with VEGF compared to unstimulated PCs. *Cacna1h* showed an increase after 48 h but not after 24 h, while *Cacna1i* showed an increase after 24 h but not after 48 h. These results show that VEGF and most likely the VEGFR-2 pathway leads to a change in T-type Ca^2+^ channels expression in PCs. There are different considerable pathways leading to this effect. Two interesting pathways are the mitogen-activated protein kinase kinase/extracellular signal-regulated protein kinases (MEK/ERK) pathway and the phosphoinositide 3-kinase/AKT/mTOR (PI3K/AKT/mTOR) pathway [62,63,64]. The study of Ferron, Capuano et al. demonstrated that *Cacna1g* was upregulated via the MEK/ERK pathway in cardio myocytes [65]. Furthermore, Wang et al. found out that the MEK/ERK pathway increased Cav3.1 expression in endothelial cells [66]. Another work could show that T-type Ca^2+^ channel blockade reduced the phosphorylation of Akt regulatory residues, threonine 308 (Thr308), and serine 473 (Ser473), leading to a decreased kinase activity, which means potential synergistic effect of VEGF and T-type Ca^2+^ channel activation [67]. The role of T-type Ca^2+^ channels in neuronal plasticity is underlined by the studies of Ly and colleagues who found that Cav3.1 is involved in the generation of LTP at the PF-PC synapse and that an irregular postsynaptic activity pattern occurs during signal transduction in Cav3.1 knockout mice [68]. In thalamic neurons, Cav3.3 and its activation also led to LTP [48]. Furthermore, Canepari (2020) hypothesised that, due to hyperpolarisation of PCs, T-type Ca^2+^ channels may induce synaptic plasticity [26]. In summary, our findings show the alteration of mRNA levels of important proteins for synaptic plasticity and neuronal signal conduction due to VEGF stimulation (Figure 8). Future experiments should focus on these effects on an electrophysiological or protein expression level and reveal regulating intracellular pathways.

## Figures and Tables

**Figure 1 cells-10-02277-f001:**
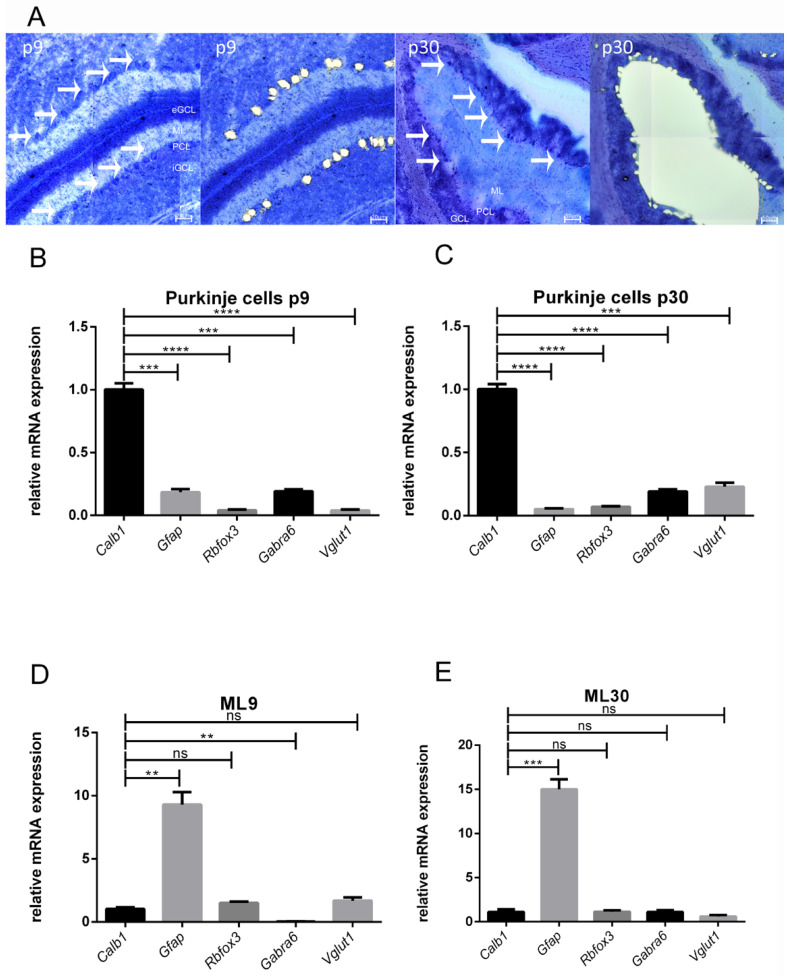
Accuracy of LMD: Laser microdissection of PCs at the age of p9 and p30 (**A**). Methylene blue-stained cryosections (12 µm) from rat cerebella. Cerebellar layers are marked as external granular cell layer (eGCL), molecular layer (ML), internal granular cell layer (iGCL) and Purkinje cell layer (PCL). White arrows indicate PCs. Scale bar: p9 = 20 µm, p30 = 50 µm (**B**,**C**) The mRNA of microdissected PCs of p9 and p30 showed high *Calb1* and low *Gfap* levels indicating only a slight contamination with glial cells. (**D**,**E**) In the microdissected molecular layer, a high expression of *Gfap* relative to markers for neurons except PCs was detected. In all four samples, the markers were normalized against *Calb1* and *Gapdh* was used as housekeeping gene. Note that 1,000,000 μm^2^ of enriched PCs and 1,000,000 μm^2^ of the ML from 20 different rat cerebella were collected. Significant differences were indicated by ** *p* < 0.01, *** *p* < 0.001, **** *p* < 0.0001; ns = not significant.

**Figure 2 cells-10-02277-f002:**
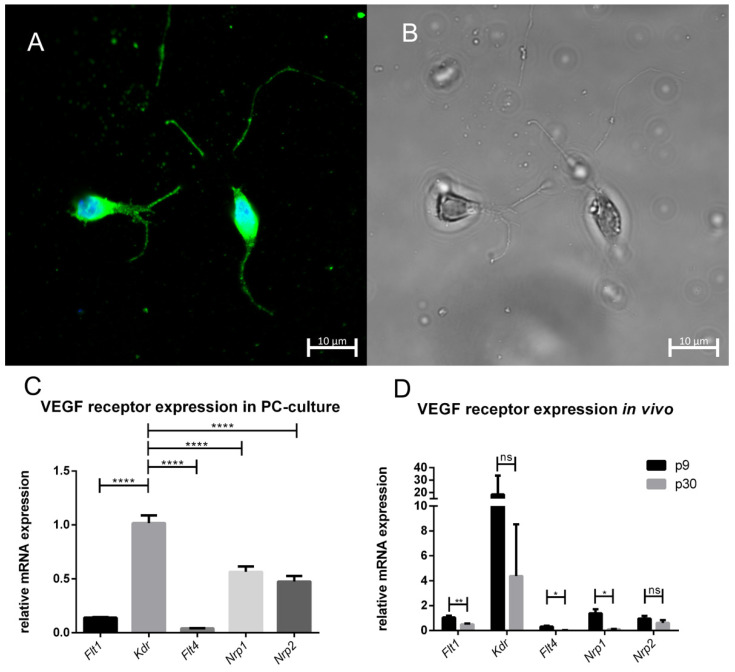
PC culture and VEGF receptor expression: (**A**) PCs were visualised via immunostaining with calbindin antibodies (green) and nuclear staining with Hoechst and in (**B**) with the help of phase-contrast optics. Scale bar: 10 µm (**C**) Quantitative analysis of VEGF receptor and co-receptor mRNA expression in an enriched PC culture. (**D**) The in vivo-grown PCs were dissected with the help of LMD. The housekeeping gene was *Gapdh* and the values were normalized against *Flt1*. (**C**): *n* = 4, (**D**): 1,000,000 μm^2^ of enriched PCs from 20 different rat cerebella were collected. The analysis was performed with the help of RT-qPCR and the 2^−ΔΔct^ method. Significant differences are indicated by * *p* < 0.05, ** *p* < 0.01, **** *p* < 0.0001; ns = not significant.

**Figure 3 cells-10-02277-f003:**
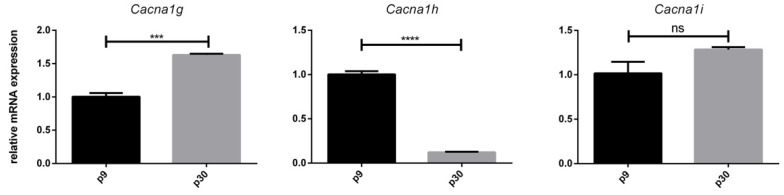
In vivo expression of *Cacna1g*, *Cacna1h*, and *Cacna1i*: Examination of the relative mRNA expression pattern of *Cacna1g*, *Cacna1h*, and *Cacna1i* in PCs was performed at p9 and p30 via qPCR of LMD samples. The values were normalized against the values of the PCs of p9, and *Gapdh* was the housekeeping gene. Note that 1,000,000 μm^2^ of enriched PCs from 20 different rat cerebella were collected. The analysis was performed with the help of the 2^−ΔΔct^ method. Significant differences were indicated by *** *p* < 0.001, **** *p* < 0.0001; ns = not significant.

**Figure 4 cells-10-02277-f004:**
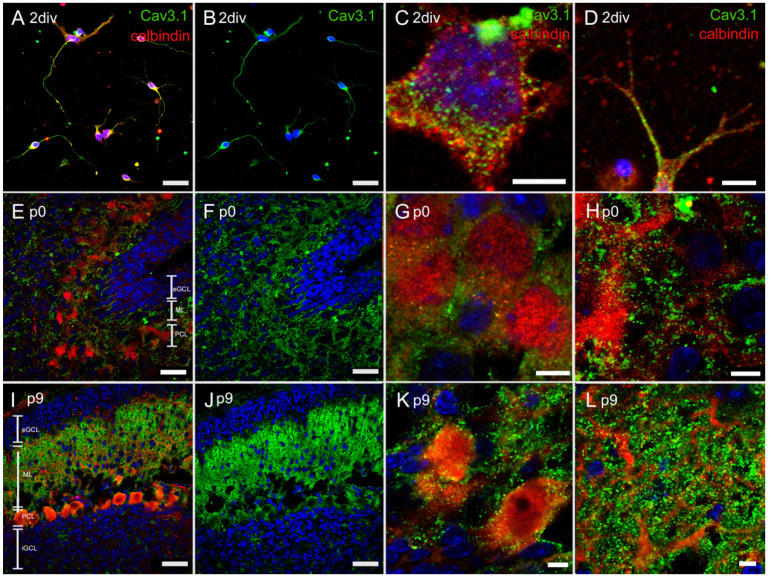
In vivo and in vitro expression of Cav3.1: Immunostaining of Cav3.1, in dissociated PCs at 2div (**A**–**D**) and cryosections of rat cerebella at p0 (**E**–**H**) and p9 (**I**–**L**). Cav3.1 (green), calbindin (red) and nuclear staining with Hoechst (blue). The cryosections showed the physiological layers of the cerebellum, namely, the external granular cell layer (eGCL), the molecular layer (ML), the PC layer (PCL), and the internal granular cell layer (iGCL). The in vitro PCs grown in serum-free medium developed dendrites or axons, with a clear Cav3.1 signal. Scale bar: 20 µm (**A**,**B**,**E**,**F**,**I**,**J**) and 5 µm (**C**,**D**,**G**,**H**,**K**,**L**).

**Figure 5 cells-10-02277-f005:**
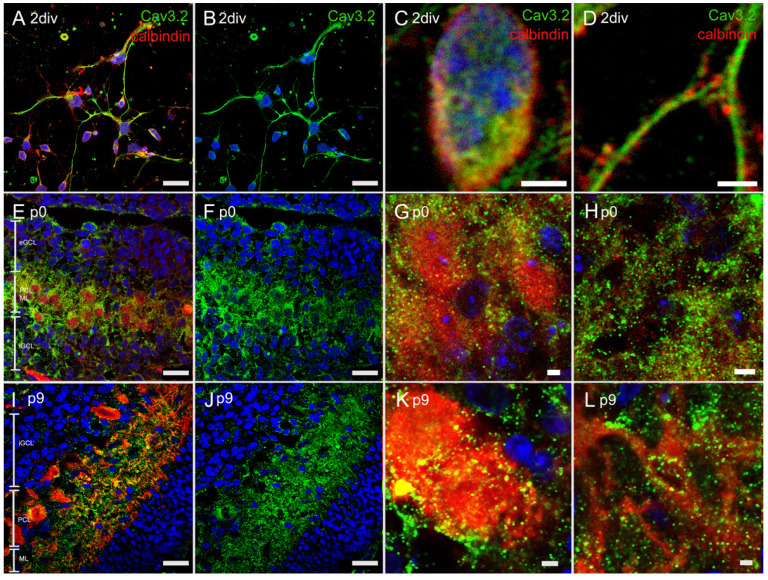
In vivo and in vitro expression of Cav3.2: Visualisation of the in vivo and in vitro distribution of Cav3.2 via anti-Cav3.2 antibodies. In dissociated PCs at 2div (**A**–**D**), Cav3.2 was prominent at the dendrites or axons, whereas in the cryosections of rat cerebella at p0 (**E**–**H**), the signal of Cav 3.2 was mostly at the soma of the PCs. In the age of p9 (**I**–**L**), there was a clear colocalization with the dendrites and the soma of the PCs. eGCL = external granular cell layer, ML = molecular layer, PCL = PC layer, iGCL = internal granular cell layer. Scale bar: 20 µm (**A**,**B**,**E**,**F**,**I**,**J**) and 2 µm (**C**,**D**,**G**,**H**,**K**,**L**).

**Figure 6 cells-10-02277-f006:**
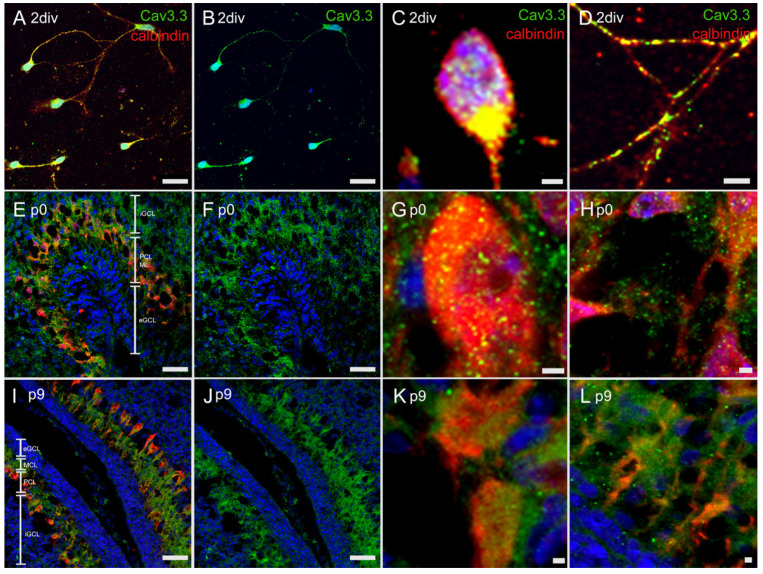
In vivo and in vitro expression of Cav3.3: Immunostaining of Cav3.3 in dissociated PCs at 2div (**A**–**D**), and in cryosections of rat cerebella at p0 (**E**–**H**) and p9 (**I**–**L**). Cav3.3 was stained green, calbindin red, and nuclear staining was performed with Hoechst (blue). In the PC culture, Cav3.3 was localized at the soma and along the dendrites or axons, whereas in cryosections of p0, cerebella staining of Cav3.3 was visible mainly near to the soma. In 9-day-old PCs, the Cav3.3-antibody strongly marked the ML, where the PC dendrites are localized. eGCL = external granular cell layer, ML = molecular layer, PCL = PC layer, iGCL = internal granular cell layer. Scale bar: 20 µm (**A**,**B**,**E**,**F**,**I**,**J**) and 2 µm (**C**,**D**,**G**,**H**,**K**,**L**).

**Figure 7 cells-10-02277-f007:**
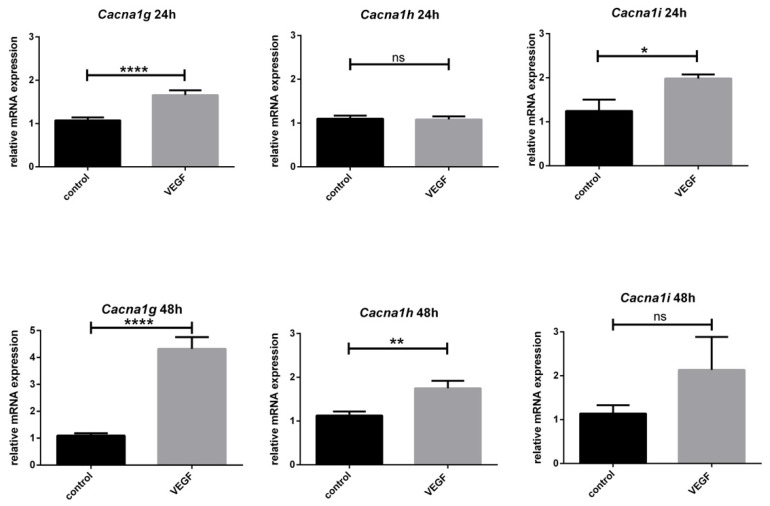
Effect of VEGF on *Cacna1g*, *Cacna1h*, and *Cacna1i* expression: The graphs show the relative mRNA expression of *Cacna1g*, *Cacna1h*, and *Cacna1i* with or without VEGF treatment for 24 h and 48 h. The values were normalised against the control group and the housekeeping gene was *Gapdh*. *n* = 8, significant differences are indicated by * *p* < 0.05, ** *p* < 0.01, **** *p* < 0.0001; ns = not significant.

**Figure 8 cells-10-02277-f008:**
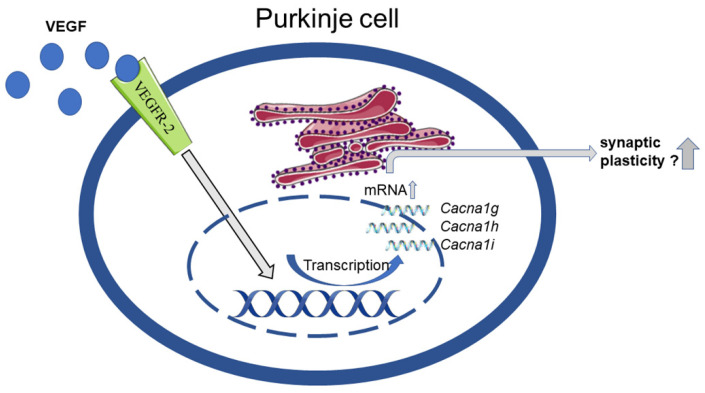
Schematic overview of the conceptualized effects of VEGF on PCs. VEGF binds to and activates VEGFR-2, followed by an intracellular signaling cascade, with consequent increase of T-type Ca^2+^ channel mRNA. It is possible that these increased levels of T-type Ca^2+^ channel mRNA enable the PC to implement synaptic plasticity more easily.

## Data Availability

All data supporting the findings of this research are available within the article.

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
