# Peer review of "Expression Pattern of T-Type Ca2+ Channels in Cerebellar Purkinje Cells after VEGF Treatment"

_cells, 2021, doi:10.3390/cells10092277_

Round 1

Reviewer 1 Report

This paper describes the expression profiles of 3 VEGF receptors and 2 co-receptors in rat Purkinje cells in culture and isolated from cerebellar sections by laser microdissection. The authors analyzed the mRNA expression levels and protein distribution of the 3 T-type Ca++ channels in the same samples and also their response to VEGF treatment.

Major comments:

This is a nice paper that initially substantiates two separate procedures for analyzing Purkinje cells (PCs) (immuno-panned PCs in culture and laser micro-dissected PCs from 12um cryosections). It is evident from the relative mRNA analysis of Calb1 and GFAP, that the subsequent analysis throughout the paper is that of PCs, and this procedure represents a good model to study PCs in general, especially in neurodegenerative models of ataxia.

Minor comments:

  1. Since the ML contains extensive arborization of PC dendrites, what is the relative level of Calb1 mRNA in the ML versus the PC layer. Fig 1B and D are both normalized to Calb1 at 1.0 so it is hard to see. This is just a point of interest with respect to calbindin synthesis in the soma and also the processes.
  2. Also, since calbindin is a Ca binding protein in the PCs, does its level change in response to VEGF treatment, when compared to the GAPDH housekeeping gene control.
  3. I am not sure if Fig 2A and B are meant to be the same as is usually done (showing ICC and phase-contrast of the same field). It represents a confusing picture as all PC in culture should be positive for Calb, if they are there in A. Clarification needed.
  4. Fig 3 should probably have statistics where the Cav mRNA levels are significantly different.
  5. Line 261; “undermined” probably not a good word to use here; likely use “demonstrate”.
  6. For all IHC figures, are there any negative controls for the primary Cav antibodies (omission of primary Ab). Low level autofluorescence is sometimes present in cerebellar sections in the 488nm range. It may be evident in Fig 4E for Cav3.1.
  7. A higher magnification of the co-staining panels would make it easier to see the overlap in the staining.

Reviewer 2 Report

Please see the attached PDF file.

Reviewer 3 Report

“Impact of VEGF on T-type-calcium-channel expression in cerebellar Purkinje-cells” is an interesting work that shows the presence of T-type-Ca2+ channels and VEGF receptors in Purkinje cells in different stages of maturation, and shows the modulation of T-type-Ca2+ channels after VEGF treatment. However, there are some critical concerns, mainly due to the experimental design and the statistical study.

Major concerns:

  • Content:

The main concern is about the statistics. There is no data from the statistical tests performed on the results shown neither in figure 1 nor in figure 2 nor in figure 3. Furthermore, the text indicates that the most expressed receptor in PCs is KDR, but Figure 2 does not show an asterisk in the graph indicating it. The same applies to the data shown in figure 1.

It is unclear why for the PCR experiments, animals of P9 and P30 are compared, while the immunohistochemistry is performed in P0 and P9 tissues, and not in P30. Please, justify it in the text.

The discussion of the results is mainly focused on the effect caused by VEGF administration in Ca2+ receptors. Too much speculation is made from those data whose only consistent result is that there is an increase in the CACNA1G channel due to the VEGF treatment.

  • Formal aspects:

Verbs in results should be written in the past form. Sometimes appear in the present, others in the past.

Minor changes:

  • Replace “Plazenta” (line 34) for placental.
  • Revise superscripts and subscript (Ca2+; H20; μm2).
  • Abbreviations must appear next to the name the first time they appear in the text. Revise LMD, PCs, GAPDH.
  • Be consistent with the punctuation of numbers.
  • Line 171, line 176 and line 181: “for 20 min respectively 30 min”. Do you mean for 20 and 30 min, respectively?
  • Line 181: “at least” should be changed to “at last”.
  • Line 196: change “especially” to “specially”.
  • Line 208: It is unclear the use of the word “qualify”.
  • Unify VEGF receptors´ names: some of them appear in the upper case, while others are written in lower cases.
  • For a better comprehension of the text, chose a form to name VEGF receptor 2 (VEGFR2 or KDR) in the text.

Round 2

Reviewer 1 Report

Thank you to the authors for addressing my questions and comments. I find the current version of the paper very good and acceptable.

Author Response

We would also like to thank the reviewer again, as we have an improved manuscript due to the helpful questions and comments.

Reviewer 2 Report

In the revised manuscript by Tjaden et al., the authors corrected many of the unacceptable typos and mislabels found in the original manuscript, as well as improving the basic English grammar in the main text. Nevertheless, several basic errors are found through the revised text, including the re-worded title. For example, the authors are strongly encouraged to reconsider their usage of hyphens (-) in key phrases such as “T-type-Ca2+-channel”, “Purkinje-cell”, and “anti-Cav3.1-antibody”.

Surprisingly, no significant new experimental data were presented in the revised manuscript. Therefore, exactly like the original manuscript, of the seven data figures presented in the revised manuscript, only Figure 7 directly addresses the effect of VEGF treatment on mRNA levels of T-type Ca2+ (CaV3) channels, followed by a schematic model in Figure 8 that speculates, without solid evidence and sufficient detail, the effect of VEGF in Purkinje cells. The other six data Figures simply serve as background characterization of protein and mRNA expression patterns of endogenous VEGF receptors and CaV3 channel isoforms in Purkinje cells, the majority of which has been reported previously in the literature. In other words, the revised manuscript still lacks basic scope in elucidating the biochemical and cell biological effects of VEGF on CaV3 channels in Purkinje cells, let alone addressing the underlying mechanisms.

My overall evaluation of the authors’ work remains the same: at its current form, this is a premature study that fails to meet the scientific scope and standard set forth by the journal Cells. 

Author Response

Thank you for your comments, we have now implemented the changes you and the editor requested regarding hyphens in the current version of the manuscript. We also understand your feedback regarding the figures, however, we believe that these are necessary attempts to emphasize our message. Partially, some aspects have already been described, as you have mentioned, but we were able to add some new aspects, which we have also discussed in the last version. All in all, in our opinion, all figures contribute to the understanding and are not yet completely described in literature. We thank you for your helpful comments which helped to improve the manuscript.

Reviewer 3 Report

The revised version of the manuscript includes the answers to all the doubts raised. Thus, the manuscript has greatly improved. The publication of the article is accepted.

Author Response

Thank you very much for this feedback, we are very pleased that we were able to address your concerns and would like to thank you again, as you also contributed to the improvement of our manuscript through your comments and questions.